# Bio-Based (Chitosan-ZnO) Nanocomposite: Synthesis, Characterization, and Its Use as Recyclable, Ecofriendly Biocatalyst for Synthesis of Thiazoles Tethered Azo Groups

**DOI:** 10.3390/polym14030386

**Published:** 2022-01-19

**Authors:** Ali H. Bashal, Sayed M. Riyadh, Walaa Alharbi, Khadijah H. Alharbi, Thoraya A. Farghaly, Khaled D. Khalil

**Affiliations:** 1Department of Chemistry, Faculty of Science, Taibah University, Al-Madinah Almunawrah 30002, Saudi Arabia; abishil@taibahu.edu.sa (A.H.B.); riyadh1993@hotmail.com (S.M.R.); 2Department of Chemistry, Faculty of Science, Cairo University, Giza 12613, Egypt; thoraya-f@cu.edu.eg; 3Department of Chemistry, Science and Arts College, Rabigh Campus, King Abdulaziz University, Jeddah 21589, Saudi Arabia; wnhalharbe@kau.edu.sa (W.A.); khalharbe@kau.edu.sa (K.H.A.); 4Department of Chemistry, Faculty of Science, Taibah University, Yanbu 46423, Saudi Arabia

**Keywords:** chitosan, zinc oxide (ZnO), nanocomposite film, heterogeneous catalysis, thiazoles

## Abstract

In recent years, nanotechnology has become a considerable research interest in the area of preparation of nanocatalysts based on naturally occurring polysaccharides. Chitosan (CS), as a naturally occurring biodegradable and biocompatible polysaccharide, is successfully utilized as an ideal template for the immobilization of metal oxide nanoparticles. In this study, zinc oxide nanoparticles have been doped within a chitosan matrix at dissimilar weight percentages (5, 10, 15, 20, and 25 wt.% CS/ZnO) and have been fabricated by using a simple solution casting method. The prepared solutions of the nanocomposites were cast in a Petri-dish and were subsequently shaped as a thin film. After that, the structural features of the nanocomposite film have been studied by measuring the FTIR, SEM, and XRD analytical tools. FTIR spectra showed the presence of some changes in the major characteristic peaks of chitosan due to interaction with ZnO nanoparticles. In addition, SEM graphs exhibited dramatic morphology changes on the chitosan surface, which is attributed to the surface adsorption of ZnO molecules. Based on the results of the investigated organic catalytic reactions, the prepared CS/ZnO nanocomposite film (20 wt.%) could be a viable an effective, recyclable, and heterogeneous base catalyst in the synthesis of thiazoles. The results showed that the nanocomposite film is chemically stable and can be collected and reused in the investigated catalytic reactions more than three times without loss of its catalytic activity.

## 1. Introduction

With the advent of nanotechnology, intensive studies have recently focused on the development of organic synthesis using nanocatalysts to fulfil greener synthetic routes by reducing the utilization and exposure of toxic chemicals [1,2,3,4,5,6]. Heterogeneous catalysts are being utilized effectively as an efficient promotor in the fine chemicals industry because of the need for more greener and safer environment. Over recent decades, a wide range of polymers have been used as powerful matrices for nanocomposites. These include natural and artificial polymers such as chitosan, gelatin, polypropylene, polyvinyl alcohol (PVA), polylactate (PLA), PLGA (poly (lactic-co-glycolic acid)]) [7,8]. Naturally occurring polysaccharides have been preferably used as efficient stabilizers to immobilize metal oxide nanoparticles, owing to their nontoxicity, biodegradability, low cost, and unique structural properties [9,10,11]. The partially deacetylated form of chitin, chitosan, is safe, nontoxic and has antibacterial and antifungal activity. Moreover, its unique structural features, as well as the presence of the hydroxyl and amino groups, make chitosan and its derivatives an ideal template to immobilize metal oxide nanoparticles [12,13,14].

Over recent years, in the developing trend towards greener, more sustainable chemistry, zinc catalysis has become increasingly popular [15,16]. Metals with great potency and environmental viability are most often used as catalysts. Zinc has shown great promise. It plays an essential part in many organic transformations due to its abundance, feasibility, and accessibility. There are a variety of bond formation reactions where zinc and zinc oxide are used as excellent catalysts, such as C-N, C-O, and C-S and in different synthetic routes such as Reformatsky [17], Negishi [18], Fukuyama [19], etc. Unfortunately, the difficult recovery and reusability of the catalyst limit its usefulness. Since the products could not be quantitatively separated and they are difficult to purify. Metal oxide nanoparticles, when added to biopolymers, can lead to nanocomposites that exhibit significantly improved properties compared to the individual polymers [20]. This has led to extensive efforts to design and synthesize these hybrid nanocomposites using simple processes. Considering their catalytic potency as a catalyst for performing chemical reactions, many researchers carefully studied their unique properties for various applications in materials science. [21,22]. Due to these factors, the problem has been addressed by doping ZnO in chitosan to produce a heterogeneous, stable, recyclable base catalyst capable of improving some investigated organic reactions. Continuing our work in heterogeneous nanocatalysis [1,2,3,23,24], a heterogeneous basic catalyst based on ZnO nanoparticles immobilized on chitosan (Figure 1) was employed in this study to synthesize thiazole-tethered azo groups.

## 2. Results and Discussion

### 2.1. Preparation and Characterization of CS/ZnO Nanocomposite Film

The simple coprecipitation method was used to prepare Chitosan-ZnO (CS/ZnO) nanocomposite films containing chitosan as a biostabiliser [24,25]. The films were also characterized by FTIR, SEM, EDX, and XRD tools.

#### 2.1.1. FTIR Characterization

The comparative FTIR spectra of the unmodified chitosan (A), ZnO nanoparticles (B), and 20 wt.% of Chitosan-ZnO nanocomposite (C) were depicted and studied and are presented in Figure 2. The absorption peaks shown in Figure 2A are those of the original chitosan, which are the combined peaks at υ = 3368 cm^−1^ of the NH2 and the OH group stretching vibrations. The absorption peaks at 2912, 2874 cm^−1^ are attributed to the asymmetric stretching of (C–H bond; CH_2_, CH_3_ groups). In addition, other absorption peaks appeared at 1602 cm^−1^ (amide carbonyl groups), 1381 cm^−1^ (CH_2_ groups), and 1057 cm^−1^ (C-O) [26]. Furthermore, Figure 2B presents the main characteristic bands of the ZnO nanoparticles at 1539, 1432, 1003, and 782 cm^−1^ [27].

For comparison, Figure 2C illustrates mixed bands composed of chitosan nanoparticles in combination with the ZnO nanoparticle structure. The reason for this appears to be the interaction between ZnO molecules and chitosan’s OH and NH_2_ groups. The characteristic peaks of ZnO are shifted to a lower wavenumber. Additionally, the wider peak at 3262 cm^−1^ of the NH_2_ and OH groups stretching vibrations, moved to a lower value, which indicates and confirms the chemical interaction between ZnO molecules and the binding sites of chitosan.

#### 2.1.2. SEM and Morphological Changes

In this study, scanning electron micrographs were used to observe the morphological changes induced by the incorporation of ZnO molecules into the polymer matrix of the chitosan surface. The SEM images of the pure chitosan (A), ZnO nanoparticles (B), and that of their hybrid nanocomposite film (20 wt.%) (C) are given in Figure 3. A dramatic morphological change in the fibrous surface of chitosan (Figure 3A) can be observed, where ZnO nanoparticles of different size and shape (circular, and elliptical) are homogeneously distributed over the surface of chitosan. The shape of chitosan-ZnO nanocomposite surface (20 wt.%) is dendritic floc-like structures in contrast to the homogeneous fibrous shape of the unmodified chitosan. Moreover, an energy dispersive X-ray (EDX) of the CS/ZnO nanocomposite film showed the presence of ZnO nanoparticles within the polymer matrix (Figure 4). The EDX profile was performed on a single spot on the spherical particles on the coating. As Zn peaks were shown in the spectra, it proves that the ZnO nanoparticles were successfully incorporated within the polymer matrix. Furthermore, the elemental analysis of the nanocomposite found that the ZnO content was ~20 wt.%.

#### 2.1.3. X-ray Diffraction

An analysis of the crystallinity and nanostructures of both native chitosan and chitosan-ZnO thin film nanocomposites was conducted (20 wt.%), using X-Ray diffraction measurements. A common characteristic of unmodified chitosan (A) can be seen in Figure 5, wherein the broad peak at 2θ = 20° is the main characteristic indicating the presence of crystalline and amorphous regions [26]. There were also additional peaks in the pattern that attested to the presence of an impurity in chitosan. According to published literature, the XRD patterns of ZnO nanoparticles (B) show peaks at 32.1°, 34.2°, and 36.4° [27]. For the hybrid nanocomposite film (20 wt.% CS/ZnO) (C), a combination of peaks appeared; however, the intensity of those spots suggests that a chemical interaction takes place between ZnO and the chitosan OH and NH2 groups. The average grain size was calculated from the XRD patterns using Debye-Scherrer formula [28].
$$D\left(\mathrm{nm}\right)=\frac{-0.9*}{cos\theta }$$
where *D* (nm) is the crystalline size in nm, λ is the wavelength of Cu-kα1 = 1.54060 A°, *β* can be calculated for the most intense peak for CS/ZnO nanocomposite pattern. The average particle size was found to be 39.6 nm.

### 2.2. CS/ZnO Nanocomposite Film as Basic Catalyst in Synthesis of Thiazole Derivatives

Initially, we investigated cyclo-condensation of 2-{1-[4-(2,4-dihydroxyphenylazo)phenyl]ethylidene}thiosemicarbazide (**1**) with appropriate *α*-keto hydrazonoyl halides (**2**), using Cs/ZnO nanocomposite as an ecofriendly basic catalyst under thermal conditions, but our trial was unsuccessful. We investigated another alternative approach for this cyclo-condensation using microwave irradiation instead of conventional heating. To estimate the optimized conditions and the proper catalyst loading of the nanocomposite, we conducted the model reaction between thiosemicarbazone **1** with *N*-phenyl 2-oxopropanehydrazonyl chloride (**2a**) in dioxane in presence of CS/ZnO nanocomposite under microwave irradiation (Figure 1 and Table 1).

The proper catalyst loading for the model reaction was carried out using 5, 10, 15, 20, 25, and 30 wt.% of nanocomposite films under the same conditions. From the results, the catalyst loading of 20 wt.% was found to be the optimal quantity of the catalyst for the maximum progress of the reaction (92% yield) after irradiation for 30 min (Table 1). In addition, the recovered catalyst was successfully used an extra five times without significant change in its catalytic potency (Figure 6).

The **3a**–**g**, groups refer to the Thiazoles tethered arylazo. Specifically, these are 2-{2-[1-(4-(2,4-dihydroxyphenylazo)phenyl)ethylidene]hydrazono}-4-methyl(aryl)-5-arylazothiazole. They were synthesized by irradiating a mixture of thiosemicarbazone **1** and *N*-aryl 2-oxopropanehydrazonyl chlorides (**2a**–**2e**) or 2-oxo-*N*,2-diaryl acetohydrazonoyl bromides (**2f,2g**) in dioxane and a basic catalyst (triethylamine or ZnO nanoparticles or CS/ZnO nanocomposite), in a comparable yield (Figure 2, see Table 2 and Table 3).

As shown in Table 3, the CS/ZnO nanocomposite is able to act efficiently as a basic catalyst for the cyclo-condensation reaction due to its synergistic effect and high yields of products for **3a**–**g** in comparison to triethylamine or ZnO nanoparticles under similar employed conditions.

The reactions depicted in Figure 2 are promoted by the presence of a basic catalyst (CS/ZnO nanocomposite), which initiates the nucleophilic displacement of the halogen atom in a-keto hydrazonoyl halides by thiol group to afford intermediate A. Dehydration of intermediate A furnished the isolable thiazole derivatives **3a**–**g** (Figure 2).

## 3. Experimental

### 3.1. Materials, Instruments, and Methods

We obtained the chitosan from Sigma Aldrich Company with the following specifications: powder, shrimp shells, product No. C3646, density = 0.15–0.3 g/cm^3^. The same company also supplied us with ZnO nanopowder (<50 nm, product No. 677450). Fourier transform infrared (FTIR) spectra of the nanocomposite were measured using potassium bromide discs in the Pye-Unicam SP300 instrument. Samples were diagnosed by high-resolution scanning electron microscopy (SEM) (HRSEM, JSM 6510A, Jeol Ltd., Tokyo, Japan) which was operated at an acceleration voltage of 20 kV in a back-scattered electron mode. Additionally, a Philips diffractometer (Model: X’Pert-Pro MPD; Philips, now PANaytical, Malvern, Worcestershire, United Kingdom) was used to acquire XRD spectra at room temperature utilizing the Cu Kα radiation (λ = 0.154 nm), from a wide-focus copper tube operating at 40 keV. The Gallenkamp capillary electrothermal device (Lister, United Kingdom) was used to determine the melting points. Microwave experiments were carried out with a CEM Discover Labmate system.

### 3.2. Preparation of CS/ZnO Nanocomposite Film

A 2 wt.% solution of medium molecular weight chitosan was formulated by dissolving in a 2% (*w/v*) aqueous acetic acid solution.

At room temperature, the resulting solution was then stirred for 48 h. Once a viscous solution had formed, it was filtered, leaving a clear, homogeneous chitosan solution, and a small portion of this solution was transferred into a 50 mL glass beaker. After vigorous stirring for 24 h, 5, 10, 15 and 20 (*w/v*%) of ZnO were added in part.

Three days of drying were needed at 50 °C in a vacuum oven to dry the solution applied to a Teflon dish (8 cm). Sodium hydroxide with 5 mL of 1M was used to help neutralize the compound to exfoliate the chitosan-ZnO nanolayer from the Petri dish and wash it with distilled water. The resulting layer was placed at room temperature in a vacuum desiccator for two days.

### 3.3. Reactions of 2-{1-[4-(2,4-Dihydroxyphenylazo)Phenyl]Ethylidene} Thiosemicarbazide (***1***) with α-Keto Hydrazonoyl Halides ***2a***–***g***

Method A: A mixture of 2-{1-[4-(2,4-dihydroxyphenylazo) phenyl]ethylidene}thiosemicarbazide (**1**) [29] (0.329 g, 1 mmol) and appropriate *α*-keto hydrazonoyl halides **2a**–**g** [30,31,32,33] (1 mmol for each) in dioxane (30 mL) was prepared and subsequently treated with five drops of triethylamine as a base catalyst. The reaction mixture was refluxed until all the starting materials were completely consumed (within 3 h. as monitored by TLC). After completion, the solvent was evaporated under reduced pressure and the residue was triturated with methanol. The precipitate was filtered, washed with methanol and crystallized from ethanol to produce compounds **3a**–**g** [29].

Method B: A mixture of 2-{1-[4-(2,4-dihydroxyphenylazo) phenyl]ethylidene}thiosemicarbazide (**1**) (0.329 g, 1 mmol) and appropriate *α*-keto hydrazonoyl halides **2a**–**g** (1 mmol for each) in dioxane (30 mL) was prepared and subsequently treated with five drops of triethylamine as a base catalyst. The reaction mixture was irradiated by MW at 300 Watt in a closed Teflon vessel until all the starting material was consumed (30–40 min as monitored by TLC). The solvent was evaporated, and the residue was triturated with methanol. The precipitate was filtered, washed with methanol and crystallized from ethanol to produce compounds **3a**–**g**.

Method C: The same procedure followed in method B, but using ZnO nanoparticles or CS/ZnO nanocomposite film (20 wt.%) instead of triethylamine. Once the reaction was completed, the nanoparticles or the film was carefully removed by filtration and washed with ethanol. The nanocomposite film can be reused multiple times in other reactions.

## 4. Conclusions

We successfully used FTIR, FESEM and EDX sample characterization methods to study chitosan−ZnO nanocomposites (as an environmentally friendly biocatalyst). In the 20 wt.% sample, we obtained an average ZnO particle size of around 30–32 nm. Our results can be compared with that of triethylamine as a conventional catalyst, where the hybrid nanocomposite film worked well as a heterogeneous catalyst for the synthesis of thiazole derivatives. Our compound also had an improved environmental effect, as the chitosan−ZnO nanocomposite was a more effective catalyst in these reactions than Triethylamine. Considering the catalytic properties, the synergistic effect resulting from the combination of the fundamental nature of both ZnO nanoparticles and chitosan itself was the main reason for the superior catalytic efficacy of the chitosan−ZnO nanocomposite. In terms of its environmentally friendly effect, the catalytic nano-film can be easily removed, recovered and reused without loss of its catalytic activity. Finally, the chitosan-metal oxide hybrid nanocomposite is a promising hybrid nanocomposite, and more research is needed on different organic transformations.

## Data Availability

The data presented in this study are available on request from the corresponding author.

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
