# Peer review of "Bio-Based (Chitosan-ZnO) Nanocomposite: Synthesis, Characterization, and Its Use as Recyclable, Ecofriendly Biocatalyst for Synthesis of Thiazoles Tethered Azo Groups"

_polymers, 2022, doi:10.3390/polym14030386_

Round 1
Reviewer 1 Report
Do to the lack of novelty of the work proposed I suggest to significantly improve the paper before publishing it.
Some major revision should be made to this work.
Following some suggestions:
- It seems that references 4 and 5 have been swapped places.
- For the first sentence references should be improved; just as an example https://doi.org/10.1016/j.ultsonch.2017.07.046 and https://doi.org/10.1016/j.mcat.2017.12.015 could be cited, with several further works.
- Some reference should be added also to the sentence in lines 41-42.
- If spectra in Figure 2 have been shifted to not overlap them, the ordinate labels should be hidden. Moreover, the spectrum for ZnO nanoparticles is presumed to be more likely the one reported as C rather than B (as also is reported in the suggested reference). Consequently, revise the text in lines 83-84.
- I suggest to substitute reference 23 in line 84 with a more appropriate one.
- The method used to evaluate the ZnO weight percentage into the film by EDX analysis should be clarified and, if EDX analysis are reported in the text, more details must be added into the experimental part of the paper. For an easiest and more precise evaluation of the amount of filler effectively added I suggest to perform a thermogravimetric analysis in oxidative atmosphere.
- Please choose a better reference than 22 in line 117: the spectra reported into the referred paper refers to different materials then your and consequently also spectra are different by yours. Moreover, if you talk about further peaks (as in line 118) they should be reported in figure. Please add the full spectra and insert the present Figure 5 as an inset.
- As the equation reported in line 125 contains only one parameter to be defined the sentence following could be written as “where D(nm) is the crystalline size….”
- I did not found in the text or in Schemes/Tables an explicit description of structure/nomenclature for molecules 2a-g and 3a-g. Please fill this void.
- As reported, Table 3 inspires that all the data reported in the table have been collected from reference 25, also the ones for the catalyst proposed as novelty in this paper. I suggest to rearrange columns or headings.
Author Response
|
|
Reviewer 1 Comments |
Our Reply |
|
1 |
Do to the lack of novelty of the work proposed I suggest to significantly improve the paper before publishing it. |
Thank you very much for the suggestion of the reviewer, some modifications, & improvements were done upon the reviewers’ comments. Novelty and aim of the work are highlighted in the introduction part. |
|
2 |
· It seems that references 4 and 5 have been swapped places. |
Thank you for your comment, the references are swapped correctly. |
|
3 |
· For the first sentence references should be improved; just as an example https://doi.org/10.1016/j.ultsonch.2017.07.046 and https://doi.org/10.1016/j.mcat.2017.12.015 could be cited, with several further works. · Some reference should be added also to the sentence in lines 41-42. |
The suggested references are added to the introduction part and more references are cited to the sentence in line 41-42. |
|
4 |
· If spectra in Figure 2 have been shifted to not overlap them, the ordinate labels should be hidden. Moreover, the spectrum for ZnO nanoparticles is presumed to be more likely the one reported as C rather than B (as also is reported in the suggested reference). Consequently, revise the text in lines 83-84. I suggest to substitute reference 23 in line 84 with a more appropriate one. |
In fig. 2, the two spectra 2B and 2C are similar but not identical due to the different finger print region that possesses a similar pattern to the chitosan. Also, the figure 2B, exhibited the characteristics of ZnO as reported in mentioned reference. The reference is replaced with similar one. |
|
|
· The method used to evaluate the ZnO weight percentage into the film by EDX analysis should be clarified and, if EDX analysis are reported in the text, more details must be added into the experimental part of the paper. For an easiest and more precise evaluation of the amount of filler effectively added I suggest to perform a thermogravimetric analysis in oxidative atmosphere. |
Additional information about the EDX measurements is added to the experimental and discussion sections. Unfortunately, the TGA apparatus is under maintenance at this time. |
|
|
· Please choose a better reference than 22 in line 117: the spectra reported into the referred paper refers to different materials then your and consequently also spectra are different by yours. Moreover, if you talk about further peaks (as in line 118) they should be reported in figure. Please add the full spectra and insert the present Figure 5 as an inset. |
Thank you for the reviewer comment, the reference is replaced by better one. |
|
|
· As the equation reported in line 125 contains only one parameter to be defined the sentence following could be written as “where D(nm) is the crystalline size….” |
The sentence is modified. |
|
|
· I did not find in the text or in schemes/Tables an explicit description of structure/ nomenclature for molecules 2a-g and 3a-g. Please fill this void. |
Thank you for your comment, names of the mentioned compounds are added. |
|
|
· As reported, Table 3 inspires that all the data reported in the table have been collected from reference 25, also the ones for the catalyst proposed as novelty in this paper. I suggest to rearrange columns or headings. |
The data are divided into two tables (Table 2 & 3) in order to two show the novelty that achieved by the invented nano catalyst. |

Reviewer 2 Report
-In the introduction part, the novelty of the work should be highlighted.
-Also, the meaning of the work should be more deeply explain. What was the scientific problem?
-How was the size of ZnO nanoparticles measured? It should be given TEM images or size distribution from DLS measurement.
-The experiments (data in Table 1) were preformed once or repeated 3-4 times. For better accuracy, the standard deviation should be given.
Author Response
Authors would like to thank the reviewer for his valuable comments.
Please see the following table of our responses, corrections and explanation based on the reviewer comments.
|
|
Reviewer 2 Comments |
Our Reply |
|
1 |
In the introduction part, the novelty of the work should be highlighted. |
The problem to be solved and target of the work is highlighted in the introduction part. |
|
2 |
Also, the meaning of the work should be more deeply explain. What was the scientific problem? |
Now in the introduction part, the two sentences, highlighted in red, explained the scientific problem and how we could solve it by the invented nanocatalyst. |
|
3 |
How was the size of ZnO nanoparticles measured? It should be given TEM images or size distribution from DLS measurement. |
Unfortunately, TEM instrument is not working now and under maintenance, so the sentence is deleted (line 106), and the size of ZnO nanoparticles is based on the application of Debye-Scherrer formula in XRD, line 132, which was found 39.6 nm. |
|
4 |
The experiments (data in Table 1) were preformed once or repeated 3-4 times. For better accuracy, the standard deviation should be given |
As usual in all of our published papers, each experiment was repeated twice only and the reported value of yield is an average of them. |

Reviewer 3 Report
The manuscript of Bashal et al. is a solid piece of work that deserves publication in Polymers, notwithstanding I personally believe that this work is better suited for a catalysis journal like Catalysts.
Before the publication, the authors should perform real recycling experiments, at least 5-10 cycles and not only 3 cycles. These results could be better illustrated through histograms rather than tables(table 2)
Author Response
Authors would like to thank the reviewer for his valuable comments.
Please see the following table of our responses, corrections and explanation based on the reviewer comments.
|
|
Reviewer 3 Comments |
Our Reply |
|
1 |
Before the publication, the authors should perform real recycling experiments, at least 5-10 cycles and not only 3 cycles. These results could be better illustrated through histograms rather than tables (table 2) |
Thanks a lot for the author, the experiments are extended to 5 trials and the table is replaced by histogram |

Round 2
Reviewer 1 Report
I am very sorry but my opinion about the deep lack of novelty of this paper can not be changed just by slightly improve the text. I persevere in my decision taking account that authors not fully addressed to the presented concerns, and some figures have already been published in previous papers of the same authors (https://doi.org/10.1016/j.jscs.2021.101276).
Also, please check tables 2 and 3: which is the catalyst used in reference 25?
Author Response
|
|
Reviewer 1 Comments |
Our Reply |
|
1 |
I am very sorry but my opinion about the deep lack of novelty of this paper can not be changed just by slightly improve the text. I persevere in my decision taking account that authors not fully addressed to the presented concerns, and some figures have already been published in previous papers of the same authors (https://doi.org/10.1016/j.jscs.2021.101276). |
The similarity in obtained analytical data for CS-ZnO and CS-CuO is attributed to the use of the same polymer matrix as a support for the metal oxide nanoparticles. However, their catalytic efficiency is totally different. Herein, the novelty of the present article is based on the use of ZnO nanoparticles (not CuO) that was incorporated within the same type of natural polymer and consequently the type of investigated catalytic organic transformations. The findings are totally different form the published study in the mentioned article where the utilized CuO nanoparticles showed a catalytic potency against different investigated organic syntheses. Zn O nanoparticles have been cited as basic catalyst in many heterocyclic syntheses [15-19]. But the difficult product purification and recovering of catalyst limited its use. Thus, the novelty here is distributed between the preparation of CS-ZnO nanocomposite and its use as a powerful heterogeneous catalyst in the selected reactions. |
|
2 |
Also, please check tables 2 and 3: which is the catalyst used in reference 25? |
Upon request of one of the reviewers, Table 2 is divided into Table 2 (for TEA) and Table 3 (for the nano catalysts). The catalyst used in this reference (25 and now 29) was triethylamine and it is studied separately in Table 2 under different conditions while the ZnO nanoparticles and CS-ZnO nanocomposite were studied in Table 3 for comparison. |

Round 3
Reviewer 1 Report
ok